# Genes Encoding Structurally Conserved Serpins in the Wheat Genome: Identification and Expression Profiles during Plant Development and Abiotic and Biotic Stress

**DOI:** 10.3390/ijms24032707

**Published:** 2023-01-31

**Authors:** Chongmei Dong, Ting-Chun Huang, Thomas H. Roberts

**Affiliations:** 1School of Life and Environmental Sciences, University of Sydney, Camperdown, NSW 2006, Australia; 2Plant Breeding Institute, University of Sydney, Cobbitty, NSW 2570, Australia

**Keywords:** wheat, serpin, expression profile, disease response, abiotic stress response

## Abstract

Serpins constitute a family of proteins with a very wide distribution in nature. Serpins have a well-conserved tertiary structure enabling irreversible protease inhibition or other specific biochemical functions. We examined the 189 putative wheat serpin genes previously identified by Benbow et al. (2019) via analysis of gene annotations (RefSeq v1.0) and combined our previous examinations of wheat ESTs and the 454 genome assembly. We found that 81 of the 189 putative serpin genes, plus two manually annotated genes, encode full-length, structurally conserved serpins. Expression of these serpin genes during wheat development and disease/abiotic stress responses was analysed using a publicly available RNAseq database. Results showed that the wheat LR serpins, homologous to Arabidopsis AtSerpin1 and barley BSZx, are ubiquitously expressed across all tissues throughout the wheat lifecycle, whereas the expression of other wheat serpin genes is tissue-specific, including expression only in the grain, only in the root, and only in the anther and microspore. Nine serpin genes were upregulated in both biotic and abiotic responses. Two genes in particular were highly expressed during disease and abiotic challenges. Our findings provide valuable information for further functional study of the wheat serpins, which in turn may lead to their application as molecular markers in wheat breeding.

## 1. Introduction

Benbow et al. (2019) identified 189 putative serpin genes in the bread wheat genome [1]. Serpins comprise a large family of proteins found in all domains of life (i.e., archaea, eubacteria, and eukarya), as well as in specific viruses [2,3,4]. Serpins have a well-conserved tertiary structure including eight to nine alpha-helices (A–H), three beta-sheets (A–C), and a reactive centre loop (RCL), which together require approximately 330–600 (typically 400) amino acid residues (aa) to form the complete polypeptide [5,6,7]. Most serpins, including those in plants, are irreversible inhibitors of serine or cysteine proteases, while other serpins have evolved to take on distinct biochemical roles that do not involve protease inhibition, such as the transport of steroid hormones in mammals [8,9,10].

During the inhibition process, a cognate protease recognises and binds to the specific amino acid residues at the reactive centre of the RCL. The consecutive P1-P1’ residues, by definition, represent the primary protease cleavage site, with the residues N-terminal to P1 known as P2, P3, etc., and those C-terminal to P1’ known as P2’, P3’, etc. The cleavage of the RCL by a protease induces a conformational change in the serpin from a metastable conformation to a relaxed, low-energy state. This conformational transition traps the covalent serpin–protease complex, inhibiting the protease irreversibly [11]. For a serpin to function biochemically as a protease inhibitor, the RCL must be 17 (rarely 16) residues in length, with glutamic acid at P17 (i.e., 17 residues N-terminal to the cleavage site) [12]. A complete RCL is a key part of the structure of a functional serpin, particularly for serpins that mediate irreversible suicide inhibition of target proteases.

Plant serpins were first identified in beer (derived from barley malt) and named collectively as protein Z [13]. Since then, serpins or their genes have been found throughout the plant kingdom, as well as in green algae [14,15]. A substantial number of serpins have been isolated or cloned from grains of Triticeae cereals, including barley, oats, rye, and wheat. Many of these serpins exhibit protease inhibition in vitro [16,17,18,19,20,21,22,23]. Plant serpins appear to play an important role in defence against plant pathogens and are known to modulate the activity of endogenous proteases. A pumpkin (*Cucurbita maxima*) serpin, CmPS-1, in the phloem sap may play a role in defence against exogenous proteases [24]. Arabidopsis AtSerpin1 negatively regulates stress-induced cell death by inhibiting the vascular cysteine protease RD21 [25,26]. Sorghum serpins were found to inhibit trypsin-like proteases present in the midgut of sorghum insect pests [27].

Cereal grain serpins constitute as much as several percent of the total protein, ranging (for example) from 1.4 to 4.0% among eight barley varieties tested [28]. They may be important for grain development and the protection of storage proteins from digestion by insects or fungi [22,29]. A non-inhibitory function of barley grain serpin Z4 (BSZ4) has been described in which the interaction between Z4 and β-amylase promotes β-amylase’s enzymatic activity and further prevents β-amylase aggregation during oxidative stress [30]. Given the functional diversity of animal serpins [5,8,9,10], it is reasonable to speculate that plant serpins may have a range of different functions that will be uncovered upon further research.

The hexaploid genome of bread wheat (*Triticum aestivum*) is the result of polyploidisation between the tetraploid *Triticum turgidum* (AABB) and the diploid *Aegilops tauschii* (DD), which gives the diploid subgenomes A, B, and D [31,32]. Although these three subgenomes share similar sequences and are highly collinear, rapid alteration of the genome landscape during evolution has resulted in gene duplication, leading to neofunctionalization, as well as gene decay, leading to gene silencing. This can generate pseudogenes and contribute to differential expression among homeologous genes [33,34]. In addition, Ramírez-González et al. [34] suggested that differential expression is subjected to transposable elements within promotor regions, epigenetic modification, or tissue-specific expression.

Identifying serpin expression patterns across different wheat tissues and growing stages can lead to the identification of genes that may be important for enhanced resistance against pathogens or tolerance to environmental stresses, constituting valuable information for wheat breeders. Benbow et al. [1] identified 189 putative serpin genes in the wheat genome [35]; however, some of the serpin genes listed did not appear to meet specific criteria for the putative gene products to be functional serpins. Here we examined these 189 putative serpin genes in detail by revisiting the wheat genome annotation [35], integrating our previous research on wheat serpins through wheat ESTs and the Chinese Spring 454-pyrosequencing database [36], and correcting specific gene annotations. We were able to refine the serpin gene complement in wheat and explore expression profiles of these genes in relation to wheat development, as well as responses to pathogens and abiotic stress.

## 2. Results

### 2.1. Serpin Identification

The availability of the wheat genome sequence makes it possible to identify all serpins in wheat. By using a hidden Markov model (HMM) for the serpin family (PFAM: PF00079) and searching with HMMER software, Benbow et al. [1] identified 189 putative serpin genes in the wheat genome. To us, it appeared likely, however, that a very substantial subset of these genes did not encode functional serpins, as the number of serpin genes identified was very large compared to the number identified in other plant genomes (e.g., the barley (diploid) genome contains 25 serpin genes, and *Brachypodium* contains 27 [15]), even taking into account the hexaploid genome of wheat. Previously, we identified 57 full-length putative serpin genes in wheat by using wheat ESTs and the Chinese Spring 454-pyrosequencing database [36] (unpublished). Together with our earlier results and annotated wheat serpin sequences downloaded from EnsemblePlants [37], we used the conservation of predicted serpin structure, domain identity, and RCL sequence as criteria to identify serpins in the wheat genome. Many predicted proteins with a small size (<330 aa), with a lack of or non-standard RCL, or with putative functions other than those established for serpins (such as protein kinase domain-containing proteins and HTH myb-type domain-containing proteins) were excluded. The detailed results of this analysis are presented in Appendix A.

We also found errors in the annotated serpins, including splice sites that were not conserved as well as intron sizes below the minimum (such as 2 bp or 4 bp). These incorrectly annotated serpins were also excluded. Some annotations were corrected (Appendix A). The final list of wheat genes encoding putatively functional serpins comprises 83 genes (Appendix A).

Of the 83 genes encoding putatively functional serpins in wheat, the B subgenome has the greatest number (32) compared to the A subgenome (24) and the D subgenome (27), which is consistent with a previous report of higher gene content in the B subgenome than in A and D [35], possibly due to genome expansion during the polyploidisation of wheat. Chromosomes 4 and 5 harbour half of the wheat serpin genes (41 out of 83), while chromosomes 1 and 7 harbour the lowest number of serpin genes (Table 1).

### 2.2. Phylogenetic Analysis

The wheat serpin amino acid sequences were aligned and a phylogenetic analysis was conducted in MEGAX. The results show that the 83 serpin genes can be separated into three major clades (I, II, and III) (Figure 1). The most highly expressed serpin genes, which include the grain serpin genes, *TaSZ1*, *TaSZ2*, and *TaSZ3* (TraesCS5A02G359700, TraesCS5B02G362000, TraesCS5D02G368900, TraesCS5A02G417800, TraesCS5B02G419900, TraesCS5D02G425800, TraesCS7D02G172000), as well as *TaSZx*, are found within Clade I (Figure 1). The name *TaSZx* is used for the wheat homeologous genes TraesCS4A02G235700, TraesCS4B02G079200, and TraesCS4D02G078000 because they are homologues of the barley Zx gene (*BSZx*) [19]. They are also the orthologs of Arabidopsis *AtSerpin1* [38]. All Zx members are known as LR serpins [14]. The reactive centre P2-P1’ Leu-Arg-X (where X is a small residue, such as Gly or Ser) is present in at least one serpin in all plant species examined, including mosses, gymnosperms, and flowering plants [14,39]. The LR serpins from barley (BSZx) and from Arabidopsis (AtSerpin1) are the most well-studied plant serpins. The phylogenetic tree indicates that the highly expressed grain-specific serpins may have evolved from the Zx+Zy (TraesCS4A02G235900, TraesCS4B02G079100) group. Most serpin genes in Clade I are expressed during wheat development.

Serpin genes in Clade II are located on chromosomes 1, 2, 5, and 6, with lower expression levels during wheat development compared to those in Clade I (Figure 1). Interestingly, the serpin genes in one particular group, which are from chromosomes 1 and 6 (TraesCS1A02G289500, TraesCS1B02G299000, TraesCS6B02G152500, TraesCS6D02G114700), were found to be expressed exclusively in the root (details in Section 2.4.2). Serpin genes in Clade III are on several different chromosomes; many are associated with no evidence for expression, whereas some are expressed in specific tissues (such as root and microspores) and under specific conditions (details in Section 2.4.2).

### 2.3. Genome Distribution and Gene Duplication of the Wheat Serpin Gene Complement

Following analysis of the serpin gene distribution in the wheat genome (Table 1), we performed an analysis of the homeologous genes based on the homologous alignment within the wheat serpin family, the distribution of each gene on homeologous chromosomes, and clustering in the phylogenetic tree. The serpin gene chromosome location and corresponding homeologues are shown in Figure 2 and Appendix A. Of the 83 serpin genes, 30 (36%) were found to belong to homeologue sets with a 1:1:1 pattern, i.e., a single gene copy per subgenome (A, B, D). Nine (11%) serpin genes have an imbalanced pattern, 1:2:1 and 3:1:1 (A:B:D configuration), meaning that there are additional paralogous genes on one subgenome resulting from a tandem duplication or a segmental duplication. A further 25 serpin genes (30%) belong to sets in which there was the loss of one homeologue, giving a pattern of 0:1:1, 1:0:1, or 1:1:0, with one group giving a pattern of 0:2:1 due to gene duplication (Appendix A). Of the remaining 19 serpin genes, some lost two homeologous counterparts due to gene silencing or mutations, e.g., TraesCS7D02G172000 (*TaSZ3*) lost its homeologues TraesCS7A02G171200ps and TraesCS7B02G076300ps (ps = pseudogene) due to mutations; others may have arisen from gene duplication/genome expansion and functional innovation as seen in the B subgenome, which has a higher number of serpin genes without homeologous counterparts.

We found that the Zy group is possibly the result of tandem duplication of the Zx group, as the genes belonging to these groups (TraesCS4A02G235700 = Zx-4A, TraesCS4A02G235900 = Zy-4A, TraesCS4B02G079200 = Zx-4B, TraesCS4B02G079100 = Zy-4B) are located adjacently or closely to each other. The K_a_/K_s_ ratio between Zx-4A and Zy-4A is 0.1834, and between Zx-4B and Zy-4B it is 0.1923 (Appendix A), implying stabilising selection. A group of serpin genes on chromosome 6 (TraesCS6A02G042000, TraesCS6A02G042100, TraesCS6A02G042200; TraesCS6B02G057800, TraesCS6B02G058100; TraesCS6D02G048100, TraesCS6D02G048700, TraesCS6D02G048800) are also located close to each other, possibly resulting from gene duplication. The K_a_/K_s_ values among these genes range from 0.3052 to 0.5552 (Appendix A).

TraesCS7D02G172000 encodes a serpin previously characterised as WSZ3 (TaSZ3), which is highly expressed in the developing grain [22,41]. However, no homeologous copies of this gene on chromosomes 7A and 7B were detected when grain protein extracts from wheat cv. Chinese Spring nulli-tetrasomic lines were examined via SDS-PAGE (unpublished). From a detailed analysis of ESTs and the wheat genome sequence, we identified that the copy of TraesCS7D02G172000 on 7A has a mutation at the splice site, such that the gene is expressed but not translated. The copy on 7B has an 8-bp insertion in the first exon, which causes a frame shift and then a premature STOP codon. This copy is also expressed but not translated into protein. In wheat RefSeq v1.0, TraesCS7A02G171200.1 and TraesCS7B02G076300.2 are the truncated serpin fragment homologues of TraesCS7D02G172000. We therefore included TraesCS7A02G171200ps and TraesCS7B02G076300ps in Figure 2. Wu et al. [41] identified the polymorphic mobility of WSZ3 in native PAGE. We identified a single nucleotide change from G65 (genotype WSZ3b) to A65 (genotype with WSZ3bʹ), which led to an amino acid residue change from Arg22 to His22, giving relatively low-mobility and high-mobility polypeptides, respectively (not shown). Arg has a pI value of 10.76, whereas His has a pI of 7.59, which explains this difference in mobility in native PAGE [41].

We found that another serpin gene, TraesCS2A02G036700 on chromosome 2A (at the edge of Clade III, Figure 1), did not have its homeologous copies on 2B and 2D annotated. We manually annotated and included these genes, as TraesCS2B02G050210* and TraesCS2D02G036000* (Appendix A), in Figure 1 and Figure 2. Chromosome 4A/5A/7B translocation in bread wheat is well known [42]; thus, homologous genes TraesCS5A02G490400 and TraesCS4D02G318300 possibly resulted from a translocation of the homeologue on chromosome 4A to 5A (Figure 2).

### 2.4. Expression Profile of Wheat Serpin Genes

We analysed the wheat serpin gene expression profile in expVIP [40], combined with our previous research on serpin expression. We also referred to expression data in Expression Altas [43] and Genevestigator [44]. A heat map of the serpin expression pattern during wheat development and biotic and abiotic stress is shown in Figure 3 and Figure 4 and Appendix A.

About 31% of the wheat serpin genes (26 out of 83) are not expressed or expressed at levels too low to be detected (expVIP). More than half of the non-expressed serpins are in Clade III of the phylogenetic tree (Figure 1). Chromosomes 3B and 2D harbour a high percentage of non-expressed serpins, probably due to a loss of activity/function during gene duplication and gene family expansion. When these 26 non-expressed serpin genes were examined in Genevestigator, most of them were found to be expressed in reproductive organs, such as the microspore, anther, and stamen, at a lower level (Appendix A).

#### 2.4.1. Highly Expressed Grain-Specific Serpins and the Correction of the TraesCS5B02G362000 Sequence

Serpins in wheat grain were previously named as wheat serpin Z proteins (WSZ or TaSZ) [16,17,22]. Of the 83 wheat serpin genes we identified, 38 are expressed in the spike and grain tissues. Only genes encoding members of the WSZ1 subfamily, WSZ2 subfamily, and WSZ3 subfamily are expressed endosperm-specifically at levels orders of magnitude higher than those of the other serpins. These genes are very poorly expressed elsewhere in the plant. Wheat grain has been shown to contain a high serpin content of up to several percent of the total protein, and their possible role as a defensive shield to protect storage proteins from digestion by insects or fungi has been proposed, partly based on the similarity of the serpin reactive centre sequences to the highly repetitive sequences of the prolamin storage proteins [20,22,29].

TraesCS5B02G362000 encodes a polypeptide of only 264 aa, which represents the C-terminal two-thirds of the WSZ1a polypeptide but lacks the N-terminal region and, thus, does not meet the structural criteria for a serpin. As WSZ1a is highly expressed, many ESTs of this gene can be retrieved and assembled into a full-length transcript sequence. In wheat RefSeq v1.0, we identified another gene, TraesCS5B02G545000LC.1, which encodes the N-terminal third of WSZ1a (97 aa). To investigate what went wrong in the annotation of TraesCS5B02G362000, we designed PCR primers according to the sequence of TraesCS5B02G545000LC.1 (forward primer) and TraesCS5B02G362000 (reverse primer). A PCR product of approximately 1.5 kb was obtained when wheat cv. Chinese Spring DNA was used as a template. The sequencing of the PCR product showed there was a mononucleotide repeat in the intron immediately following the splice site near the 5′-end of the intron (Appendix A), which hindered genome sequencing and annotation. We manually corrected the sequence of TraesCS5B02G362000 so that it represented the true WSZ1a gene (TraesCS5B02G362000*, Figure 5). We also corrected an error in the original WSZ1a sequence, where there was a 4-aa difference due to a “G” insertion at position 341 and a “C” deletion at 352, as encoded by the cDNA sequence (Z49890) (Appendix A). The expression data showed that TraesCS5B02G362000 is polymorphic; this gene is not expressed in wheat cv. Azhurnaya but is expressed in Chinese Spring. Other research showed a WSZ1a presence/absence polymorphism in a large collection of wheat cultivars [41], which was found to be associated with wheat milling yield [45].

All members of the wheat grain serpin group are highly and exclusively expressed in the grain, except TraesCS5B02G419900, which is also expressed in root, sheath, and flag leaf at a lower level and is upregulated by phosphate starvation.

#### 2.4.2. Serpin Genes Expressed Ubiquitously and Tissue-Specifically

Genes for the Zx subfamily, TraesCS4A02G235700, TraesCS4B02G079200, and TraesCS4D02G078000, are the second group of highly expressed serpin genes in wheat. Moreover, they are the only group expressed in all tissues through the whole lifecycle (Figure 3), which is similar to the near-ubiquitous expression of genes encoding BSZx [46] and AtSerpin1 [14,25,47] in barley and Arabidopsis, respectively. Zx is an LR serpin, and its orthologous gene appears to be present in all plants. Its function in the regulation of programmed cell death, as established in Arabidopsis and rice [26,48], is likely to be shared among all plants.

The Zy subfamily genes, TraesCS4A02G235900 and TraesCS4B02G079100, although closely related to Zx, are expressed at lower levels than the Zx genes. TraesCS4A02G235900 is expressed in the root, seedling, vegetative stem, and developing grain, but its expression is not detected in leaves, the reproductive stem, flag leaf, or spike (Figure 3). The reactive centre sequences of the Zy serpins are highly similar to those of the Zx subfamily (which have P2-P1’ LRS): TraesCS4A02G235900’s P2-P1’ has evolved to LES, and TraesCS4B02G079100 to LMS. The negatively charged P1 Glu and hydrophobic P1 Met in the two Zy serpins are in stark contrast to the positively charged P1 Arg in Zx. Therefore, the products of the active Zy genes are expected to have inhibitory specificities very different from those of the Zx serpins. Another branch in this group, Zz (TraesCS5A02G490400), is expressed at an even lower level than the Zy genes (Figure 3).

Other wheat serpin genes in Clade I, which are on chromosomes 2, 3, and 4, are all expressed at relatively low levels in various tissues, including developing grains, compared to the Zx subfamily and the Z1 + 2 + 3 group. The two members of one group, TraesCS4A02G205200 and TraesCS4D02G106100, are expressed in the root, seedling, leaf, stem, and spike, but not in the grain (Figure 3). These serpins are putatively non-inhibitory, as evidenced by an examination of the sequence of the RCL hinge region, and were named as Z9 in rice [49]. Orthologous genes of Z9 were identified in other cereal and grass species, including barley, rice, Brachypodium, and maize [49]. Their functions are likely to be special and conserved, but they have not been elucidated.

In Clade II, a group of serpin genes located on chromosome 6, TraesCS6A02G042200 and TraesCS6D02G048800, are expressed at low levels in almost all tissues, with higher levels of expression in the outer pericarp and root tip [43,44]. Sequence analysis identified a signal peptide located at the N-terminus of the protein, indicating that the translated serpin is destined for the secretory pathway, possibly directed outside of the cell, to specific organelles, or inserted into membranes. Next to this group in the phylogenetic tree is a group of serpins from chromosome 5, which also feature a signal peptide and are expressed specifically in the anther and microspore (data from Genevestigator). Members of the next group on chromosome 1, TraesCS1A02G289500 and TraesCS1B02G299000, also encode products with a signal peptide and are exclusively expressed in the root (Figure 3). We speculate that serpins in this clade evolved to have specific functions. Another group of serpin genes on chromosome 6, TraesCS6B02G152500 and TraesCS6D02G114700, are also root-specific, although they do not encode products with a signal peptide (Figure 3). Next to this group in the phylogenetic tree is a group of serpins on chromosome 2: TraesCS2A02G369100, TraesCS2B02G386300, TraesCS2B02G386500, and TraesCS2D02G365900. Although they are not substantially expressed according to expVIP, they are specifically expressed in the anther, microspore, floret, and stamen according to Genevestigator.

On the edge of Clade III, a group of serpin genes, TraesCS4B02G309900, TraesCS4B02G310200, and TraesCS4D02G308100, are specifically expressed in the root (Figure 3). Next to this group is TraesCS2A02G036700, for which no evidence for expression is found in expVIP but the specific expression in the stamen and anther is recorded in Genevestigator. This gene has homeologous genes on chromosomes 2B and 2D, which are missed in the RefSeq v1.0 annotation. We annotated them manually as TraesCS2D02G036000* and TraesCS2B02G050210* (Appendix A). Other serpin genes in this clade are either not expressed or expressed specifically in the anther, microspore, floret, and stamen (Genevestigator) at low levels (Appendix A).

#### 2.4.3. Serpin Genes Responsive to Abiotic or Biotic Stress

Figure 4 shows the wheat serpin genes that respond to biotic and abiotic stress from the analysis of data in expVIP. Some of these genes are upregulated and some are downregulated during pathogen attacks or during environmental stress (heat, drought, cold, etc.). We found that the database Expression Altas [43] curated some experiments not recorded in expVIP, e.g., *Xanthomonas translucens* infection of wheat leaf. We therefore included serpin genes responsive to *X. translucens* in Figure 4 and Table 2.

As stated earlier, Zx serpins were found in all plant species examined and are expressed ubiquitously; thus, their function may be essential for plant life. We found a downregulation of Zx genes during *Fusarium graminearum* infection. Interestingly, an upregulation of TraesCS4A02G235700 was observed during *Zymoseptoria tritici* infection and an upregulation of TraesCS4B02G079200 during *Xanthomonas translucens* infection, as well as an upregulation of TraesCS4D02G078000 during *Blumeria graminis* infection. TraesCS4A02G235700 is also upregulated during cold and drought stress, while TraesCS4D02G078000 is upregulated during heat stress (Figure 4). The Zy serpin gene, TraesCS4A02G235900, although not recorded in expVIP for abiotic or biotic responses, is recorded as upregulated in response to drought stress in the seedling stage of roots in Genevestigator.

Two wheat serpin genes, TraesCS4A02G436000 and TraesCS2B02G530600, are highly expressed during *F. graminearum* and *X. translucens* infection; they also respond to drought and heat. Another gene, TraesCS4D02G231200, responds to stripe rust infection; it also responds to heat and drought, as well as combined drought and heat stress. TraesCS3D02G301100 is responsive to *F. graminearum, Z. tritici*, and PAMP elicitors chitin and flg22 (Figure 4, Table 2). These genes may gain interest among wheat breeders for disease resistance and drought/heat tolerance.

Most serpin genes in Clade III are specifically expressed in anthers and microspores (Appendix A). In the experiment recorded in expVIP with the cold treatment of cultured microspores, these genes are all downregulated (Figure 4).

## 3. Discussion

The availability of a high-quality wheat reference genome [35] makes it possible for a genome-wide analysis of any gene family of interest. Using bioinformatic tools and publicly available genomic and transcriptomic data, Benbow et al. [1] identified 189 putative serpin genes and studied their responses to disease [1]. This so-called ‘serpinome’ of wheat is much larger than we predicted based on the serpin complement of barley, Arabidopsis, and other species [15,47,49]. It was possible that the high-quality wheat genome sequence and annotation give a complete picture of true serpin genes in the wheat genome, but it was also possible that the large size of the serpinome is due to some misclassification. We performed a detailed analysis of the 189 genes and found more than one-third encode proteins smaller than 330 aa, which does not allow the polypeptide to fold to form a functional serpin [5,6]. Some genes on the list produce a putative protein product that lacks an RCL. Some genes encode other proteins, such as protein kinase and HTH myb-type domain-containing protein (Appendix A). We refined this serpin gene list and combined our previous analysis of annotated serpin genes with some manually corrected/annotated/assembled serpin genes to give a list of 83 serpin genes in the bread wheat genome.

Phylogenetic analysis revealed that wheat serpins can be classified into three clades. Clade I contains the highly expressed grain serpins and the Zx + y + z serpins. The amino acid identity between the wheat Zx members and the biochemically well-characterised BSZx is ~94%. Recombinant BSZx was found to be an efficient inhibitor in vitro of serine proteinases with trypsin-like specificity, including several blood coagulation factors, at P1–P1’ Arg-Ser, as well as proteinases with chymotrypsin-like specificity at the overlapping reactive centre P2–P1 Leu-Arg [18,19]. Undoubtedly, the inhibitory properties of wheat Zx are very similar to those of BSZx.

For AtSerpin1 from Arabidopsis, the inhibition of trypsin and the cysteine proteinases AtMC9 and AtMC4 has been demonstrated in vitro [38], and the dominant in vivo target protease has been identified as the papain-like cysteine protease, RESPONSIVE-TO-DESICCATION 21 (RD21) [25]. AtSerpin1 limits RD21 activity through set-point control to prevent ‘unintended’ cell death [26]. Normally, RD21 and AtSerpin1 are localized to different subcellular compartments. Upon release of RD21 from ER bodies and vacuoles into the cytoplasm, cell death is facilitated, but only if the total number of active RD21 molecules substantially exceeds the number of cognate AtSerpin1 molecules already in the cytosol. Set-point control thus refers to the “counting off” (or titration) of target protease molecules by serpins through the irreversible formation of 1:1 AtSerpin1:RD21 covalent complexes. T-DNA insertion lines lacking specific RD21 protease activity and AtSerpinHA over-expression lines showed lower levels of cell death, while insertion lines lacking AtSerpin1 displayed enhanced cell death. Thus, once the activity of RD21 overpowers the inhibitory activity of AtSerpin1, the protease can cleave its substrate proteins and facilitate programmed cell death. Whether members of the wheat Zx + Zy + Zz are functional homologues of AtSerpin1 in the regulation of cell death has not been shown; however, at least for the Zx serpins, the perfect conservation of the reactive centre suggests a highly conserved function. We speculate that the upregulation of Zx and Zy gene expression by biotic and abiotic stresses such as cold, drought, and heat (Table 2) might reflect a need to control the activity of target proteases released from organelles by the loss of integrity of their membranes.

The wheat serpin phylogenetic tree (Figure 1) indicates that the grain serpins (Z1 + Z2 + Z3) are more closely related to the Zx subfamily than to the other wheat serpin subfamilies. This relationship, combined with the apparent lack of Z1, Z2, and Z3 serpins outside the tribes Triticeae (e.g., wheat and barley) and Poeae (e.g., oat) of the family Poaceae, suggests that these subfamilies evolved from Zx serpins after the grasses evolved as a distinct family. Inhibition activities of Z1 and Z2 members have been demonstrated in earlier research [18,22]. Their abundance in mature wheat grain suggests their functions in defence/storage.

Some serpins in Clade I are worthy of attention due to their upregulated expression during biotic and abiotic stress. TraesCS4A02G436000 and TraesCS2B02G530600 are highly expressed during biotic and abiotic challenges. They both respond to *Fusarium graminearum* (Fusarium head blight) and *Xanthomonas translucens* infection, as well as heat and drought. TraesCS2B02G530600 is also reported to respond to the pathogens *Puccinia graminis* (wheat stem rust) and *Septoria tritici* (blotch) [44]. TraesCS4D02G231200 responds to *Puccinia striiformis* (wheat stripe rust) as well as to heat and drought conditions. TraesCS3D02G301100 responds to *F. graminearum* and *Septoria tritici*, as well as to drought. The non-inhibitory serpin gene TraesCS4D02G106100 is responsive to the pathogens of Fusarium head blight and bacterial leaf streak, as well as cold.

The expression profiles of the wheat serpin genes shown in this study infer functional specificity. Serpins in Clade I, apart from the grain-specific serpins, are mostly ubiquitously expressed. They respond well to biotic and abiotic stresses. Serpins in Clade II have evolved to have more specific functions. Three groups contain a signal peptide in the translated serpins, so they are possibly secreted outside of the cell or to organelles or membranes. The expression profile for serpin genes in this clade indicates a specific expression in the root or anther and microspore, except for one group of serpin genes on chromosome 6 that expressed almost ubiquitously but at a low level. Most serpin genes in Clade III exhibit a highly specific expression pattern in the anther, microspore, floret, stamen, and rachis, i.e., the reproductive organs. They may have specific functions.

Serpin genes are distributed in all seven chromosome groups, with the highest number on chromosomes 4 and 5. More than 40% of the serpin genes have homeologues on all three subgenomes. Gene duplication and gene loss are also observed. Gene duplication and gene-family expansion are important mechanisms of evolution and environmental adaptation. Gene gain was observed in hexaploid wheat when the gene number was compared to its diploid ancestors [32]. The gene-family expansion could lead to functional innovation [34,50]. On the other hand, gene duplication and hexaploidy could lead to functional redundancy, allowing the accumulation of mutations and resulting in gene loss, as we observed in the *WSZ3* gene and its homeologues. Rehman et al. [15] reported the identification of 27 and 25 serpin genes in the genomes of Brachypodium and barley (both diploids), respectively. In this study, 83 serpin genes were identified in wheat (hexaploid), which was therefore comparable to its diploid counterparts.

Wheat is one of the most important and widely cultivated crops on Earth. Global wheat production has faced yield stagnation due to increased plant stress from pathogens and extreme weather events, partly caused by climate change [51,52]. One of the most effective ways to mitigate yield losses is to increase the tolerance and resistance of wheat to abiotic and biotic stresses through breeding. However, useful stress-tolerance genes must be provided to breeders. We expect that the identification of functional serpin genes in the wheat genome will be of substantial value in this regard.

## 4. Materials and Methods

### 4.1. Reference Serpin Lists

A list of serpin genes from Benbow et al. (2019) was downloaded from FigShare (https://doi.org/10.25387/g3.7910417.v1, accessed on 15 January 2021). A list of putative wheat serpins from the wheat annotation (RefSeq v1.0) was downloaded from EnsemblePlants [37]. A list containing all reviewed wheat serpins (unpublished) from an analysis of wheat ESTs and the Wheat 454 genome assembly [36] was also included. Most genes in these three lists were in common, so the lists were combined.

### 4.2. Serpin Gene Identification

Sequences of the predicted wheat serpin genes, along with AtSerpin1, were obtained through EnsemblPlants [37] and UniprotKB [53]. Where the putative serpin genes had isoforms, all the isoform sequences were included. Based on the predicted protein names in the database, serpins were classified into “serpin domain-containing protein”, “uncharacterised protein”, and “other proteins”. Based on the length of sequences, an additional three categories (<330 aa, between 330 and 600 aa, and >600 aa) were made. An extra “unable to confirm” category was created for the putative serpins that were listed only in EnsemblPlants (not in UniProtKB). Some sequences retrieved were identified as incorrect due to annotation errors (see Section 4.4); these sequences were removed.

The serpins with sequences between 330 and 600 aa were retrieved and aligned with AtSerpin1 using Clustal Omega [54] with default settings. After the alignment, sequences were examined individually to ensure the presence of an RCL sequence. The serpins without a complete RCL sequence were filtered, and the predicted structures of the remaining serpins were examined using AtSerpin1 [25] as a reference. Any serpins with major defects, such as a lack of an alpha-helix or a beta-sheet, were excluded. Finally, the putative serpins in the final list were searched locally on BLAST [55] with blastp to compare the similarity with serpins from other species.

### 4.3. Phylogenetic Analysis

The amino acid sequences of the identified wheat serpins were aligned with Clustal Omega. Phylogenetic analysis was conducted using MEGAX [56] with 200 bootstrap replicates. The evolutionary history of wheat serpins was inferred using the maximum likelihood method and the JTT matrix-based model [57]. The phylogenetic tree was constructed based on the highest log-likelihood generated via neighbour joining and BioNJ algorithms, with a matrix of pairwise distances used to place serpins into different clades. These values were generated via MEGAX default settings [56]. The tree was then modified and graphed in iTOL [58].

### 4.4. Chromosomal Locations, Homeologous Genes, and Gene Duplication

Gene locations were provided in IWGSC RefSeq v1.0 annotation [59] and EnsemblPlants [37]. The exon/intron sites were also checked in these databases. Homeologous genes were determined based on high sequence homology (>80% identity at the protein sequence), location on the A-, B-, and D-subgenomes, and clustering in the phylogenetic tree. Gene duplications were also identified by their high homology and clustering in the phylogenetic tree. The synonymous substitution rate (K_s_), the non-synonymous substitution rate (K_a_), and the K_a_/K_s_ ratio between the duplicated gene pairs were calculated using DnaSP v6. TBtools [60] was used to construct a circus figure to show gene location and homologous genes.

### 4.5. Expression Analysis—Wheat Tissues

The dataset relating to tissue expression under the no stress (normal) condition was downloaded from expVIP [34,61] and classified according to the experiment types and growth stage. The expression data points with a transcript count (in transcripts per million, TPM) lower than 0.5 were treated as no expression (TPM = 0) and excluded. This was to ensure no false expression was recorded. The remaining data points were converted into log_2_(TPM) values. Data from the wheat cultivars Chinese Spring and Azhurnaya were included. The final data points were classified according to tissue type (root, spike, leaf, and stem) and the growing stage of the wheat (seedling, vegetative, and reproductive). A heatmap was produced in Excel according to the level of expression.

### 4.6. Expression Analysis—Abiotic and Biotic Stresses

RNAseq datasets containing wheat stress responses were downloaded from expVIP [34,61], and each data point was treated the same as in the wheat expression analysis (TPM > 0.5, log_2_ FC (Fold Change) > |±0.5|). A heatmap was generated in Excel according to the difference in log_2_ FC values between the control and stress conditions, with positive values indicating an upregulation and negative values indicating a downregulation associated with the stress.

## Figures and Tables

**Figure 1 ijms-24-02707-f001:**
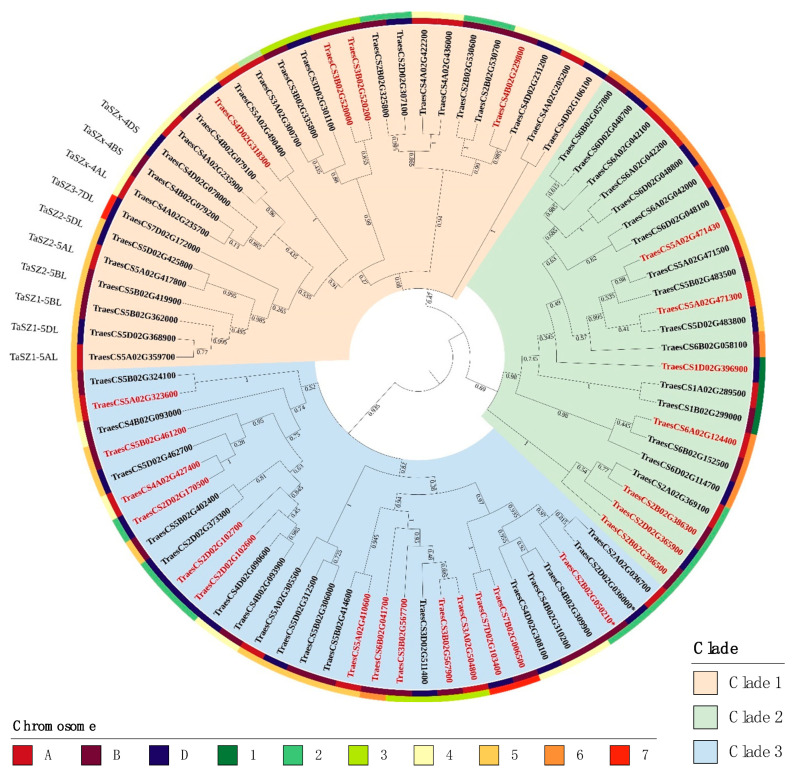
Wheat serpin phylogenetic tree based on an amino acid sequence alignment. The three clades are highlighted with different colours: beige (Clade I), green (Clade II), and blue (Clade III). The colours on the rings of the figure indicate subgenomes and chromosomes. Gene names in black indicate expression, whereas those in red indicate no expression detected in expVIP [40].

**Figure 2 ijms-24-02707-f002:**
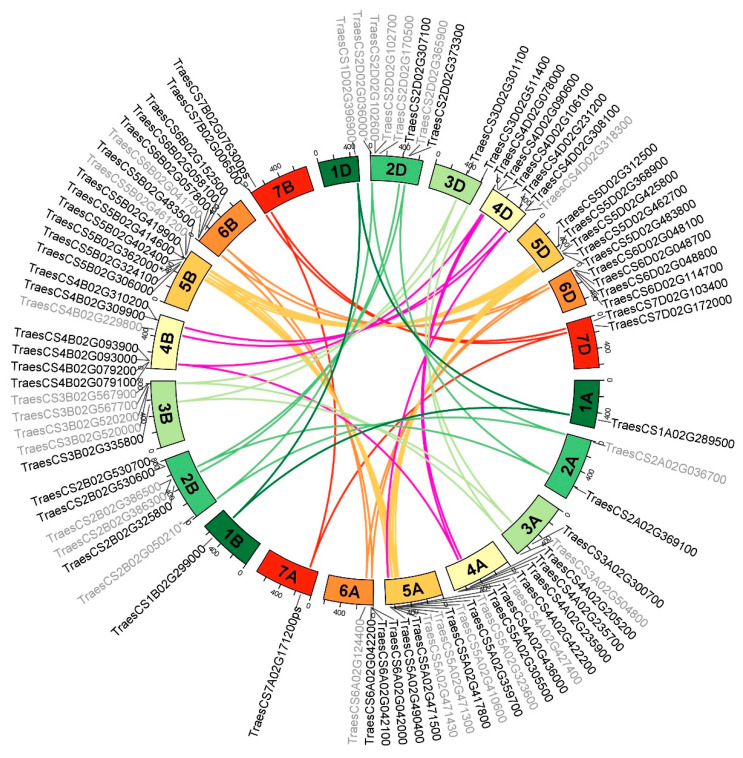
Serpin gene chromosome locations and homeologous genes in subgenomes A, B, and D shown by curved lines inside the ring. Gene names ending with * indicate that the sequence was corrected through manual annotation. Gene names ending with ps indicate pseudogenes. Names in black indicate genes that are expressed; names in grey indicate genes not expressed (according to expVIP).

**Figure 3 ijms-24-02707-f003:**
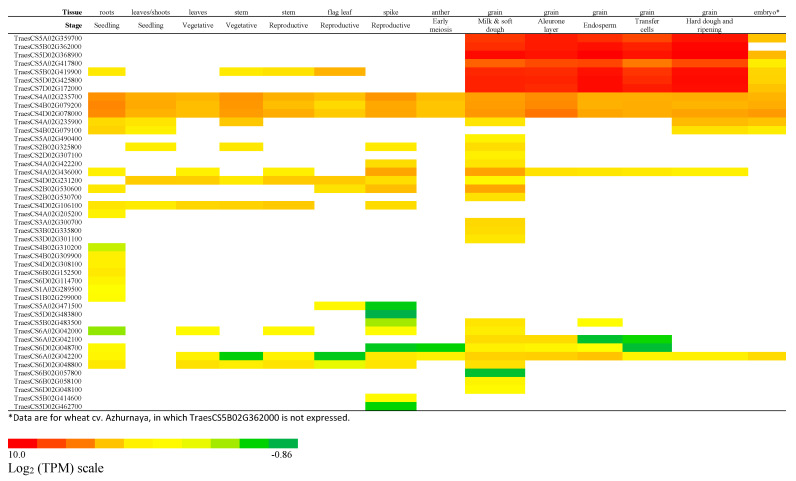
Serpin gene expression profile during the wheat lifecycle in various tissues/organs. Data are mainly from cv. Chinese Spring, except for embryo data, which are from cv. Azhurnaya in expVIP.

**Figure 4 ijms-24-02707-f004:**
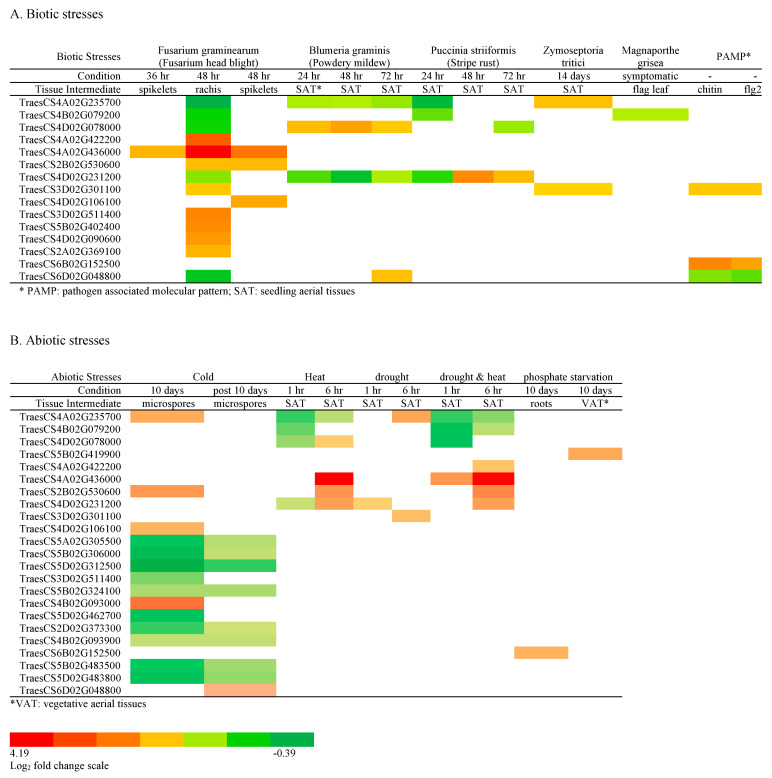
Serpin gene responses to (**A**) biotic and (**B**) abiotic stresses. Data are from expVIP, except for *Zymoseptoria tritici* treatment, which is from Expression Altas.

**Figure 5 ijms-24-02707-f005:**
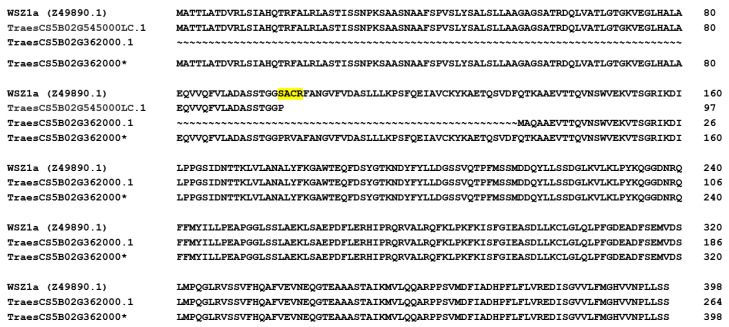
Alignment of WSZ1a with the protein products of manually annotated TraesCS5B02G362000*, TraesCS5B02G545000LC.1, and TraesCS5B02G362000.1. The yellow highlighted four amino acid residues “SACR” were a sequence error due to a base insertion and a base deletion. Detailed DNA sequences and ESTs are in Appendix A.

**Table 1 ijms-24-02707-t001:** Numbers of genes encoding putatively functional serpins found on each chromosome of the wheat genome.

Chromosome	Subgenome			
	A	B	D	Total
1	1	1	1	3
2	2	6	7	15
3	2	5	2	9
4	6	7	6	19
5	9	8	5	22
6	4	4	4	12
7	0	1	2	3
Total	**24**	**32**	**27**	**83**

**Table 2 ijms-24-02707-t002:** Genes upregulated during biotic and abiotic stress. Yellow highlighted genes are highly upregulated with log_2_FC > 2.

Clade	Wheat Serpin Gene	Biotic Stress *	Abiotic Stress
Clade I	TraesCS4A02G235700	Zt	cold, drought
	TraesCS4B02G079200	Xt	
	TraesCS4D02G078000	Bg	heat
	TraesCS5B02G419900		P-starvation
	TraesCS4A02G422200	Fg, Xt	drought and heat
	TraesCS4A02G436000	Fg, Xt	heat, drought, and heat
	TraesCS2B02G530600	Fg, Xt	cold, heat, drought, and heat
	TraesCS4D02G231200	Ps	heat, drought, drought, and heat
	TraesCS3D02G301100	Fg, Zt, chitin, flg22	drought
	TraesCS4D02G106100	Fg, Xt	cold
Clade II	TraesCS6D02G048800	Bg	
	TraesCS6B02G152500	chitin, flg22	P-starvation
	TraesCS2A02G369100	Fg	
Clade III	TraesCS3D02G511400	Fg	
	TraesCS5B02G402400	Fg	
	TraesCS4D02G090600	Fg	

* Fg = *Fusarium graminearum* (Fusarium head blight); Bg = *Blumeria graminis* (powdery mildew); Ps = *Puccinia striiformis* (stripe rust); Zt = *Zymoseptoria tritici* (*Septoria tritici* blotch); Xt = *Xanthomonas translucens* (bacterial leaf streak). Chitin and flg22 are pathogen-associated molecular pattern (PAMP) elicitors.

## Data Availability

The data presented in this study are available in the article and Appendix A.

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
