# Peer review of "Genes Encoding Structurally Conserved Serpins in the Wheat Genome: Identification and Expression Profiles during Plant Development and Abiotic and Biotic Stress"

_ijms, 2023, doi:10.3390/ijms24032707_

Round 1
Reviewer 1 Report
The authors have described the in-silico study of Genes encoding structurally conserved serpins in the wheat genome. The information provided in this manuscript will help for the exploration of the functional role of serpins genes in crop improvement.
The study is well-described, and the following are the major comments to be considered while revising this manuscript for publication.
1. The introduction section can be shortened to focus on the questions beings addressed in this study along with the details of earlier studies and significance.
2. The paralogous genes that originated at the time of duplication events led to the expansion of gene families. Therefore prediction of duplication events and Ka/Ks analysis may be addressed.
3. Protein–Protein Interaction analysis using the STRING database is required to find the interaction of serpins with other proteins.
4. qRT-PCR Analysis may be required for validation of in-silico expression results.
Author Response
Reviewer 1
The authors have described the in-silico study of Genes encoding structurally conserved serpins in the wheat genome. The information provided in this manuscript will help for the exploration of the functional role of serpins genes in crop improvement.
The study is well-described, and the following are the major comments to be considered while revising this manuscript for publication.
- The introduction section can be shortened to focus on the questions beings addressed in this study along with the details of earlier studies and significance.
The Introduction in our original manuscript was 60 lines in length, which we consider sufficiently succinct. Unfortunately, it is not possible to shorten the Introduction but also include details of earlier studies and their significance, as requested by the reviewer. Indeed, the inclusion of such information would require a lengthening of the Introduction. We added the information about our earlier results in the section 2.1.
- The paralogous genes that originated at the time of duplication events led to the expansion of gene families. Therefore prediction of duplication events and Ka/Ks analysis may be addressed.
We have now predicted recent gene duplication events by identifying pairs of wheat serpin genes in the same subgenome for which the predicted proteins have an amino acid sequence identity of at least 80%. We then conducted a Ka/Ks analysis of these pairs of genes using the program DnaSP (Supplementary Table 8). For example, the genes TraesCS2B02G386300 and TraesCS2B02G386500, which are in the B subgenome, were analysed and found to have a Ka/Ks ratio of 0.301. The Ka/Ks values obtained for the 15 pairwise comparisons made between the highly similar wheat serpin genes ranged from 0.183 to 0.589.
- Protein–Protein Interaction analysis using the STRING database is required to find the interaction of serpins with other proteins.
We thank the reviewer for this suggestion. Our manuscript is focussed on serpin genes, rather than properties of the serpin proteins. Thus, the identification of proteins that might interact with the wheat serpins does not lie within the scope of our study. We will be producing a manuscript on the wheat serpin proteins in the future, in which we can make use of the STRING database.
- qRT-PCR Analysis may be required for validation of in-silico expression results.
Our study is based on analysis of the wheat genome and hundreds of individual expression analyses, the results of which are available in several databases. The ‘in-silico’ expression results we analysed are not predictions, but rather are based on experiments conducted by others. It is not possible for us to replicate even a small subset of these transcript-based experiments and to do so would have limited value for our study.
Reviewer 2 Report
The article is well written and present novel finding on topic under discussion. I have just few comments/suggestions.
Gene/s names must be Italic.
Remove short paragraphing from Introduction and Discussion section. Introduction may have 3-4 paragraphs.
Remove outdated citations if any.
Mention the novelty statement of your article.
Author Response
Reviewer 2
The article is well written and present novel finding on topic under discussion. I have just few comments/suggestions.
Gene/s names must be Italic.
The gene names given in our manuscript are in the form of loci; e.g., TraesCS5D02G368900 indicates that this gene is from Triticum aestivum cv. Chinese Spring, that it is found in the D genome, and is on chromosome 5 in location 368900 (the 02 in the gene name is the version of the gene annotation). The convention is that gene names in this locus form are not written in italics. Note that Benbow et al. (2019) also did not use italics for these gene names. The equivalent situation in (say) Arabidopsis thaliana is that the serpin AtSerpin1 is encoded by the gene AtSerpin1 (in italics) for which its locus is At1g47710 (not in italics).
Remove short paragraphing from Introduction and Discussion section. Introduction may have 3-4 paragraphs.
Each of our paragraphs in the Introduction focussed on a distinct subtopic; however, we have now combined the paragraph on the origin of our understanding of plant serpins with the paragraph on the known functions of some of the plant serpins. The Introduction in the revised manuscript has six paragraphs, which we believe is the clearest way to the present the information. For the Discussion, the shortest paragraph is seven lines long, and we don’t believe that any increase in clarity we will be achieved by combining any of the paragraphs in the Discussion.
Remove outdated citations if any.
We believe that we should refer to original journal articles to reference particular points made in the text. There are no articles in our reference list that we consider to be outdated.
Mention the novelty statement of your article.
We have described the novelty of our study in the final paragraph of the Introduction, in which we state that, ‘We were able to refine the serpin gene complement in wheat and explore expression profiles of these genes in relation to wheat development and disease and abiotic stress responses’.
Reviewer 3 Report
Previously report by Benbow et al. (2019) identified the 189 putative wheat serpin genes. This study by Dong et al. found that 81 of these 189 putative serpin genes, plus two manually annotated genes, encode full-length, structurally conserved serpins. Then, using a publicly available RNAseq database, the author analyzed expression of these serpin genes during wheat development and disease/abiotic stress response. These findings might provide some more detailed information for further functional study of the wheat serpins. But the overall novelty of this story is low and lacks experiment to verify.
The other concern, the author excluded serpin genes that was mainly based on the size of putative serpin genes. It might be helpful to find genes that encode full-length, structurally conserved serpins. But, other small, excluded serpin genes might be also important, because some of them probably can work together to make function.
Some minor comments,
1) Line 140, Line 143, 144. When the author statement that “TraesCS4D02G308100) were also found to be expressed in roots”, the author needs to cite the reference or website or figures/tables. Similar problems were scattered in the MS.
2) Line 325, in table 2, “Genes upregulated during biotic and abiotic stress. Yellow highlighted genes are highly upregulated”. Directly show the fold change in table or describe the fold change in text when cite is better.
Author Response
Reviewer 3
Previously report by Benbow et al. (2019) identified the 189 putative wheat serpin genes. This study by Dong et al. found that 81 of these 189 putative serpin genes, plus two manually annotated genes, encode full-length, structurally conserved serpins. Then, using a publicly available RNAseq database, the author analyzed expression of these serpin genes during wheat development and disease/abiotic stress response. These findings might provide some more detailed information for further functional study of the wheat serpins. But the overall novelty of this story is low and lacks experiment to verify.
We believe that the novelty of our study is sufficient for publication as the verification of gene identity is crucial to understand the functions of the wheat serpins and the utility of their genes as molecular markers. It is vital that researchers are made aware that only a subset of putative serpin genes identified using genomic analysis software represent genes that encode full-length, functional serpins. Without our study, researchers might assume that all 189 putative serpin genes in wheat encode functional serpins, but we have shown that less than half of these genes meet the corresponding sequence criteria. The Benbow et al. (2019) was an excellent first step in identifying the serpin gene complement of wheat, but our study enables future research to take advantage of the edited serpin gene complement in wheat breeding.
The other concern, the author excluded serpin genes that was mainly based on the size of putative serpin genes. It might be helpful to find genes that encode full-length, structurally conserved serpins. But, other small, excluded serpin genes might be also important, because some of them probably can work together to make function.
There are very strict amino acid sequence criteria that define a protein as a serpin. Our study is focussed on serpins. It is possible that some of the small genes excluded in our analysis encode functional proteins, but these are definitely not serpins.
Some minor comments,
1) Line 140, Line 143, 144. When the author statement that “TraesCS4D02G308100) were also found to be expressed in roots”, the author needs to cite the reference or website or figures/tables. Similar problems were scattered in the MS.
We have now added information giving the figures/tables associated with particular points made regarding expression throughout the manuscript.
2) Line 325, in table 2, “Genes upregulated during biotic and abiotic stress. Yellow highlighted genes are highly upregulated”. Directly show the fold change in table or describe the fold change in text when cite is better.
The data in Table 2 are sourced from several databases. The expression levels are represented differently in these various databases. Moreover, the yellow highlighted genes are associated with high levels of expression in different experiments. We feel that the main point communicated in this table is clear with the current amount of information provided.
Reviewer 4 Report
The manuscript ‘Genes encoding structurally conserved serpins in the wheat genome: identification and expression profiles during plant development and abiotic and biotic stress' is a comprehensive and interesting work that involved studying the expression profile of serpin protein in different tissues of wheat and under various stress conditions.
Consider including the serpin protein structure or a schematic view of the protein in the main manuscript or supplementary material
Given the functional diversity of animal serpins, reference? Detail more about animal serpins
Please elaborate on your previous research on serpins in detail
The expression profile of wheat serpin genes in different tissues is interesting. An interpretation table would add great value to the manuscript.
The authors talk about protein mobility and PAGE analysis but there is no further information
Minor comments
In abstract - disease and abiotic challenge. What disease the authors are referring to? By abiotic challenge, whether the authors mean abiotic stress?
Plant Kingdom need not be capitalized in line 50
Line 59 Can you provide a range of percentages?
The manuscript needs to be thoroughly checked for grammatical errors
Line 95, can you provide some specific numbers from other plant genomes like rice, barley and so?
Author Response
Reviewer 4
The manuscript ‘Genes encoding structurally conserved serpins in the wheat genome: identification and expression profiles during plant development and abiotic and biotic stress' is a comprehensive and interesting work that involved studying the expression profile of serpin protein in different tissues of wheat and under various stress conditions.
Consider including the serpin protein structure or a schematic view of the protein in the main manuscript or supplementary material
The structure of serpins has been presented in many of the articles cited in our manuscript. This includes the only plant serpin for which a structure has been determined experimentally, that of AtSerpin1 from Arabidopsis. A detailed analysis of this structure is found in Lampl et al. (2010), which we cite four times in the manuscript and for which one of us (Roberts) was a co-author.
Given the functional diversity of animal serpins, reference? Detail more about animal serpins
We have cited several journal articles that discuss in detail the functional diversity of animal serpins, including Gettins (2002) and Law et al. (2006). We have now cited these two references following the statement in the Introduction, ‘Given the functional diversity of animal serpins’.
Please elaborate on your previous research on serpins in detail
We have now included some additional information on our previous (unpublished) research on the wheat serpins.
The expression profile of wheat serpin genes in different tissues is interesting. An interpretation table would add great value to the manuscript.
We are not sure what the reviewer means by ‘interpretation table’. The reviewer needs to give us more information if we are to act on this suggestion.
The authors talk about protein mobility and PAGE analysis but there is no further information
We have now provided some additional information about protein mobility and PAGE analysis in the manuscript.
Minor comments
In abstract - disease and abiotic challenge. What disease the authors are referring to? By abiotic challenge, whether the authors mean abiotic stress?
In the abstract, the diseases referred to are those for which gene expression experiments have been conducted and made available in various databases. These include diseases caused by pathogenic bacteria and fungi. The abiotic stresses referred to are also from these databases, and include heat, cold and drought. Given the word limit of the abstract, it is not possible to include this information there.
Plant Kingdom need not be capitalized in line 50
We have now removed the capitals.
Line 59 Can you provide a range of percentages?
We have now provided a range of percentages.
The manuscript needs to be thoroughly checked for grammatical errors
We have now checked the manuscript thoroughly for grammatical errors.
Line 95, can you provide some specific numbers from other plant genomes like rice, barley and so?
We have now included the number of genes encoding full-length serpins in barley and Brachypodium (25 and 27, respectively) in the sentence.
Round 2
Reviewer 1 Report
Thanks for addressing the said changes
Author Response
Thanks for reviewing our revised manuscript.
Reviewer 3 Report
The authors have satisfactorily addressed/explained for my concern. I do not have any major comments, but some minor comments. The authors need to check the formatting/font style thoroughly before publication. Some minor errors are listed below. Please check the similar errors throughout the manuscript.
1) When cite the figures and tables in text, the authors need to refer to it in order. In the revised version, the order of these citations was messy. Such as Fig. 3 was cited after Fig. 1. Supplementary Table 7 was cited/appeared at the first of all Supplementary Tables in text. The authors need to re-arrange the order for their figures and tables.
2) In text, “Arabidopsis”, the font style should be regular. The name of a specific species, such as “Arabidopsis thaliana”, will be italic. Please change it throughout the text.
3) Line 127, Gene name "TaSZ1, TaSZ2 and, TaSZ3", should be italic. Please check the similar problem throughout the text. Other full gene ID in text, such as TraesCS5A02G359700 was in regular that it’s correct.
4) Line 108, “Supplementary Table 1 and Table 2.” Change to “Supplementary Tables 1 and 2”.
5) Line 208, “in Figure 1 and 2” change to “in Figures 1 and 2”.
6) Line 225, “tree (Fig. 1.).” to “tree (Fig. 1).”
7) Line 446, “[50,34]” to “[50, 34]”.
Author Response
1) When cite the figures and tables in text, the authors need to refer to it in order. In the revised version, the order of these citations was messy. Such as Fig. 3 was cited after Fig. 1. Supplementary Table 7 was cited/appeared at the first of all Supplementary Tables in text. The authors need to re-arrange the order for their figures and tables.
We thank the reviewer for pointing this out. We have now made edits in the text to ensure that the tables and figures are cited in the correct order. This includes changing the order of the supplementary files and tables. Concerning the citations of the figures in the main manuscript, we have now made changes so that they are in order:
Figure 1 is first cited in line 124, Figure 2 in line 167, Figure 3 and Figure 4 in line 226, and Figure 5 in line 270.
For the tables in the main manuscript, Table 1 is first cited in line 118 and Table 2 in line 352.
Supplementary Tables 1 and 2 are first cited in line 106, Supplementary Table 3 in line 110, Supplementary Table 4 in line 167, Supplementary Table 5 in line 190, Supplementary Tables 6 and 7 in line 226, Supplementary Table 8 in line 234.
Supplementary File 1 is first cited in line 112, Supplementary File 2 in line 211, and Supplementary File 3 in line 268.
2) In text, “Arabidopsis”, the font style should be regular. The name of a specific species, such as “Arabidopsis thaliana”, will be italic. Please change it throughout the text.
We have now changed ‘Arabidopsis’, when it is written alone, to normal font throughout the manuscript, while maintaining ‘Arabidopsis thaliana’ in italics.
3) Line 127, Gene name "TaSZ1, TaSZ2 and, TaSZ3", should be italic. Please check the similar problem throughout the text. Other full gene ID in text, such as TraesCS5A02G359700 was in regular that it’s correct.
We have now made the suggested changes.
4) Line 108, “Supplementary Table 1 and Table 2.” Change to “Supplementary Tables 1 and 2”.
We have now made this change.
5) Line 208, “in Figure 1 and 2” change to “in Figures 1 and 2”.
We have now made this change.
6) Line 225, “tree (Fig. 1.).” to “tree (Fig. 1).”
We have now made this change.
7) Line 446, “[50,34]” to “[50, 34]”.
We have now made this change. We thank the review for these detailed comments.